# Impact of Red Complex Bacteria and TNF-α Levels on the Diabetic and Renal Status of Chronic Kidney Disease Patients in the Presence and Absence of Periodontitis

**DOI:** 10.3390/biology11030451

**Published:** 2022-03-16

**Authors:** Jaideep Mahendra, Plato Palathingal, Little Mahendra, Khalid J. Alzahrani, Hamsa Jameel Banjer, Khalaf F. Alsharif, Ibrahim Faisal Halawani, Janani Muralidharan, Pandapulaykal T. Annamalai, Shyam Sankar Verma, Vivek Sharma, Saranya Varadarajan, Shilpa Bhandi, Shankargouda Patil

**Affiliations:** 1Department of Periodontics, Meenakshi Ammal Dental College and Hospital, Meenakshi Academy of Greater Education and Research, Chennai 600095, India; janani.harini718@gmail.com; 2Department of Periodontics, PSM College of Dental Science and Research, Thrissur 680519, India; platoos@gmail.com; 3Department of Periodontics, Dean, Maktoum Bin Hamdan Dental University, Dubai 122002, United Arab Emirates; dean@mbhduc.org; 4Department of Clinical Laboratories Sciences, College of Applied Medical Sciences, Taif University, P.O. Box 11099, Taif 21944, Saudi Arabia; ak.jamaan@tu.edu.sa (K.J.A.); h.banjer@tu.edu.sa (H.J.B.); alsharif@tu.edu.sa (K.F.A.); i.halawani@tu.edu.sa (I.F.H.); 5Department of Biochemistry, Jubilee Medical College Hospital, Thrissur 680005, India; drannamala@gmail.com; 6Department of Nephrology, Jubilee Medical College Hospital, Thrissur 680005, India; shyamvarmas@gmail.com; 7Department of Periodontics, Desh Bhagat Dental College and Hospital, Mandi Gobindgarh 114141, India; colviveksharma@gmail.com; 8Department of Oral Pathology and Microbiology, Sri Venkateswara Dental College and Hospital, Chennai 600130, India; vsaranya87@gmail.com; 9Department of Restorative Dental Sciences, Division of Operative Dentistry, College of Dentistry, Jazan University, Jazan 45142, Saudi Arabia; shilpa.bhandi@gmail.com; 10Department of Maxillofacial Surgery and Diagnostic Sciences, Division of Oral Pathology, College of Dentistry, Jazan University, Jazan 45412, Saudi Arabia

**Keywords:** creatinine, cytokines, ELISA, inflammatory mediators, inflammation, periodontal disease, periodontal pathogens, red complex bacteria, tumour necrosis factor α

## Abstract

**Simple Summary:**

Periodontitis, referred to as gum disease, is a serious bacterial infection that damages the surrounding structures of the teeth, including the bones, lastly resulting in tooth loss without prompt treatment. The disease primarily occurs as a result of improper maintenance of the oral cavity, eventually causing gum infections, and teeth that are filled with food, tartar, and bacterial deposits. These harmful bacteria under the influence of various environmental and genetic risk factors may migrate to the blood vessels and reach various other organs, including the kidneys, thereby triggering the infection and resulting in the progression of the systemic diseases. In this study, we focused on identifying the levels of gum-disease-causing bacteria and their byproducts, namely, TNF alpha in patients with or without chronic kidney disease. The study comprised 120 participants categorised into 4 groups on the basis of patients who had gum disease and a kidney disorder. Gum, renal, and diabetic parameters were recorded. The bacteria and their products were assessed and found to be higher in patients having both gum disease and CKD disorder. This study indicates that patients with severe gum disease and poor oral health are at risk of developing kidney problems. Thus, the early prevention of gum ailments might reduce the risk of future kidney diseases.

**Abstract:**

Scientific evidence shows a positive association in the etiopathogenesis of periodontitis and chronic kidney disease (CKD). Various confounding factors, such as obesity, diabetes, and inflammation, also play a significant role in the progression of CKD, which remains unexplored. We hypothesise the role of red complex bacteria with various confounding factors associated with chronic kidney disease. The study comprised a total of 120 participants categorised into 4 groups: the control group (C), periodontitis subjects without CKD (P), periodontally healthy chronic kidney disease subjects (CKD), and subjects having both periodontitis and CKD (P + CKD), with 30 subjects in each group. Demographic variables, and periodontal, renal, and diabetic parameters were recorded. Tumour necrosis factor (TNF)-α levels and those of red complex bacteria such as *Prophyromonas gingivalis* (*P.g*), *Treponema denticola* (*T.d*), *and Tonerella forsythia* (*T.f*) were assessed, and the obtained results were statistically analysed. Among the various demographic variables, age showed a level of significance. Mean PI, GI, CAL, and PPD (the proportion of sites with PPD ≥ 5 mm and CAL ≥ 3 mm) were elevated in the P + CKD group. Diabetic parameters such as fasting blood sugar (FBS) and HbA1c levels were also greater in the P + CKD group. Renal parameters such as eGFR and serum creatinine levels were greater in CKD patients. The estimation of red complex periodontal pathogens such as *Pg*, *Td* and *Tf* levels were significantly greater in the P and P + CKD groups. Pearson correlation analysis revealed significant correlation of red complex bacteria with all variables. Greater levels of *P.g*, *T.d* and *T.f* were found in the P groups, thus indicating their important role in the initiation and progression of inflammation of periodontitis and CKD, with diabetes as one of the confounding factors. The study also confirmed a log-linear relationship between TNF-α levels and red complex bacteria, thereby demonstrating the role of inflammatory biomarkers in periodontal disease progression that could contribute to the development of systemic inflammation such as CKD.

## 1. Introduction

Chronic kidney disease (CKD) affects the quality of life of affected individuals. It can easily progress to advanced stages, thereby leading to greater morbidity and mortality rates. CKD is more prevalent in Asian populations in comparison with other ethnic groups. [1]. The disease affects 8% to 13% of the total global population [2]. The principal aetiology of mortality in individuals suffering from CKD is the severity of the condition with increased inflammation that leads to cardiac complications [3].

Periodontitis, on the other hand, is an inflammatory condition caused by Gram-negative bacteria associated with both local and systemic immune responses that lead to the destruction of the periodontium [4]. Periodontal disease results from a plethora of mechanisms associated with the cross-link between host factors and microbes [5,6]. Periodontitis on progression leads to intense local synthesis of proinflammatory cytokines that can enter the bloodstream, thereby resulting in elevated levels of inflammatory biomarkers in both gingival tissue and serum [7,8]. Thus periodontal disease is a major infectious scaffold that could potentiate systemic inflammatory levels, increasing patient morbidity [9]. In order to promptly diagnose and treat periodontal diseases, their classification is mandatory to both help in organising healthcare to patients, and assists scientists and researchers in understanding the aetiopathogenesis and treatment of such diseases in a sequential meticulous pattern. There are various classification systems proposed for the categorisation of these periodontal diseases, of which the newly accepted 2017 AAP classification of periodontal and peri-implant diseases is applicable on categorising the disease on both the basis of its severity, and of the presence or absence of risk factors and complexity [5]. 

Periodontitis is a common and alterable risk factor for CKD, where it is hypothesised to cause renal impairment through an inflammatory pathway [10]. Inflammation stimulates the invasion of periodontal microorganisms in the host tissue, and directly or indirectly enters the circulatory system [11]. An increased number of inflammatory mediators alter renal parameters, which in turn affects renal functions, thereby leading to the progression of chronic kidney disease. Greater serum antibody levels to periodontal pathogens reflect their systemic dissemination, thereby resulting in their vascular and hepatic activation [12].There are various studies that discuss the role of periodontitis in CKD, but its impact in the presence of various confounding factors in the progression of the disease still remains unexplored [13,14].

The present research was conducted to assess the role of demographic and periodontal parameters, TNF-α, and red complex bacteria such as *P.g*, *T.d* and *T.f* in association with confounding factors such as GFR, serum creatinine, FBS and HbA1c levels on chronic kidney disease individuals in the presence or absence of periodontitis. The first null hypothesis of the current study is that there are no significant differences when comparing the levels of TNF-α and red complex bacteria in chronic periodontitis patients with or without chronic kidney disease. The second null hypothesis of this study is that there is no influence of diabetic status in these patients with periodontitis with or without chronic kidney disease.

## 2. Materials and Methods

### 2.1. Study Design

A cross-sectional study was performed following STROBE guidelines to report observational studies. The sample size was calculated using G-Power software to be 120 subjects with a power of 95%. The current study was conducted between January 2019 and February 2020 in Thrissur, Kerala, India, where patients were enrolled from PSM College of Dental Science and Research, Thrissur, and Thrissur Dialysis Center, India. Among the 180 examined subjects, 30 volunteers had come for a master health check-up in the general hospital of PSM College of Dental Science and Research, Thrissur, were both periodontally and systemically healthy, and were categorised as the control group (C). We enrolled 40 periodontitis patients without CKD from the PSM Dental College of Sciences, Kerala, India, of which 10 patients were excluded as they had other systemic conditions. Lastly, 30 patients were selected with glomerular filtration rate > 90 mL/min, determined in accordance with the classification of CKD_EPI 2009, and hence were grouped as the P group. We recruited 43 periodontally healthy patients, diagnosed with CKD (Stages 2–4) from Thrissur Dialysis Center, India. Among them, 13 were excluded for having other systemic conditions and adverse habits such as smoking. Hence, 30 patients were selected for the CKD group. We recruited 47 periodontitis subjects with CKD from Thrissur Dialysis Center, India, and 17 patients of which were excluded as they did not consent to be part of the study, thereby leading to a total number of 30 patients in the CKD + P group (Figure 1). Following a detailed explanation of the research protocol to the study participants, we obtained informed consent prior to the study being conducted. The Institutional Ethical Review Board of PSM College of Dental Science and Research (Thrissur, India) approved the research proposal with protocol no. PSMDC/IEC/01/2015. All procedures were executed following the Declaration of Helsinki of 1975, as revised in 2013. The confidence intervals were calculated with retrieved records from the hospital, which were 95% and 80%, with a final sample of 120 individuals. 

Lastly, the study comprised four groups: C group: 30 systemically and periodontally healthy subjects; P group: 30 systemically healthy periodontitis subjects; CKD group: 30 periodontally healthy subjects with CKD; P + CKD group: 30 CKD patients with periodontitis. Inclusion criteria: (a) willingness to participate; (b) age range: 35–65 years; (c) individuals with at least 12 natural teeth in the oral cavity; (d) in the case of CKD and C groups, subjects with intact periodontal health were recruited; (f) in the case of the P and CKD + P groups, individuals with periodontitis in accordance to the American Academy of Periodontology and European Federation of Periodontology 2017 classification were recruited, where interdental CAL was detectable at >2 nonadjacent teeth with CAL ≥ 3 mm and probing pocket depth >3 mm present in >2 teeth [15]. 

Chronic kidney disease was divided into five stages in accordance with the classification in Kidney Disease: Improving Global Outcomes (KDIGO) 2009, where predialytic patients were recruited for the study on the basis of the determined glomerular filtration rate (eGFR). Participants with GFR greater than or equal to 90 mL/min/1.73 m^2^ accounted for the C group. The CKD group comprised individuals clinically diagnosed with renal failure with a GFR range from 89 to 15 mL/min, and in predialysis conservative management [16]. 

Exclusion criteria: (a) individuals who had undergone periodontal treatment within six months prior to the study; (b) subjects with smoking, tobacco, and alcohol habits; (c) subjects with other systemic diseases such as cardiovascular diseases, hypertension, sarcoidosis, tuberculosis, carcinoma, rheumatoid arthritis, and immunosuppressive conditions, and subjects with other inflammatory conditions; (g) individuals with eGFR < 15 mL/min/1.73 m^2^ (there was a chance that they would be so ill that they could not participate in the study or would have commenced dialysis). 

### 2.2. Sample Collection

#### 2.2.1. Blood Collection

Venous blood samples were collected by venepuncture in the morning after a 12 h fasting period in 5 mL centrifuge tubes without anticoagulant. Collected samples were left to clot for 1 h at room temperature, and the serum was then centrifuged for a period of 20 min at 3000 rpm for a period of 10 min. Centrifuged sera were stored at −70 °C until analysis. 

#### 2.2.2. Subgingival Plaque Sample Collection

On the first visit of the patient, during initial clinical examination, samples of plaque from a subgingival area were collected using Gracey curettes (Hu-Friedy, Chicago, IL, USA) from the deepest periodontal sites of periodontitis subjects in the presence and absence of chronic kidney disease. Collected plaque samples were mixed with RNA later solution (Qiagen, Hilden, Germany) for the prevention of transcriptome degradation, and stored at −80 °C till further experimentation. Following subgingival plaque collection, patients were subjected to an oral prophylaxis procedure for the removal of local factors.

### 2.3. Assessed Study Parameters

#### 2.3.1. Demographic Variables

Demographic information collected from the hospital records of the patients was (1) age, (2) gender, (3) body mass index, and (4) socioeconomic status.

#### 2.3.2. Periodontal Parameters

One trained blinded investigator used a William periodontal probe to perform intraoral examination of the periodontium. Periodontal parameters were PI [17], GI [17]^,^ PPD, and CAL. Recordings were noted to the closest mm. The severity of periodontitis was graded and staged. Periodontal diagnosis of the selected subjects was in accordance with the guidelines presented by the 2017 International Workshop for the Classification of Periodontal Diseases and Conditions of the American Academy of Periodontology and the European Federation of Periodontology [16].

#### 2.3.3. Diabetic Assessment

Diabetic parameters such as HbA1C levels and fasting blood sugar levels were recorded for all participants. 

#### 2.3.4. Renal Parameters

Serum creatinine values were determined using an automated method from the collected blood samples with a semiautomated biochemical analyser. Estimated glomerular filtration rate (eGFR) was calculated with the serum creatinine values using the CKD-EPI equation of 2009. 

#### 2.3.5. Tumour Necrosis Factor-α

The concentration of inflammatory biomarker TNF-α was analysed using the ELISA method through the analysis of the venous blood samples collected from subjects of all four groups. Human TNF alpha ELISA kit (ab181421) was used to determine the marker by adhering to the instructions provided by the manufacturer. Prior to analysis, standardisation was performed, and samples were maintained to room temperature. The concentration of TNF-α levels was measured in picograms per millimetre (pg/mL).

### 2.4. Molecular Analysis

#### RT-PCR Analysis

Subgingival plaque samples collected from the subjects were subjected to RT-PCR for the detection and quantification of red complex bacteria (*P.g*, *T.d* and *T.f*). Genomic DNA was isolated from the samples by using a rapid DNA kit (QIAamp DNA Minikit (QIAGEN Inc., 9300 Germantown Road, Germantown, MD 20874, USA)) by adhering to the protocol provided by the manufacturer. These extracted DNA samples were quantified using a spectrophotometer, where further complementary DNA (c-DNA) synthesis and RT-PCR analysis were performed using the Stratagene MX3000P (Agilent Technologies, 5301 Stevens Creek Blvd., Santa Clara, CA, USA). The amplification of c-DNA samples was performed in triplicate to eliminate errors, adhering to the standard operating procedure provided by the Takara kit (TB Green TMP remix Ex Taq TMII PCR Kit; Takara Bio Inc., Shiga, Japan).

Double-standard DNA binding dye SYBR Green I (KAPA SYBR FAST qPCR Kit) using species-specific primers was utilised in the detection of *Prophyromoans gingivalis*, *Treponema denticola*, and *Tonerella forsythia*, and reaction efficacy was optimised. Melt curve analysis was conducted in every sample for identification of contaminants, multiple amplicons, and nonspecific products.

PCR products were visualised with 2% agarose gel electrophoresis and 100 bp molecular marker DNA with ethidium bromide control. Gene quantity was calculated with the comparative cycle threshold unit (CT) method. CT was used for the expression of the number of red complex bacteria, and CT values were inversely proportional to bacterial counts. 

### 2.5. Statistical Analysis

Statistical Package for Social Science (SPSS, (IBM Corporation, Chicago, IL, USA) software program version 17) for Microsoft Windows was used to statistically analyse the obtained data.

Normal distribution of data was observed. Descriptive statistics were presented with the aid of numbers and percentages. Mean and standard deviation for every parameter (demographic, renal, and periodontal) was determined for every group, and the level of significance was determined. To observe significant differences among groups, ANOVA was used. Pearson correlation coefficient analysis was performed to correlate the red complex bacteria with all the other parameters. It was considered to be statistically significant if *p* value ≤ 0.05. 

## 3. Results

Mean age was significantly greater in P + CKD (61.47 + 10.99) as compared to that in the other groups. Other demographic variables did not reach the level of statistical significance among the groups. Male subjects had more prevalence in the CKD and P + CKD groups as compared to females, but they were insignificant (Table 1).

Mean PI (2.27 + 0.23), GI (2.42 + 0.18), PPD (3.18 + 0.24), CAL (1.14 + 0.48), average proportion of sites with PPD ≥ 5 mm (21.87 + 7.49), and mean proportion of sites with CAL with ≥ 3 mm (1.14 + 0.4) were greater in the P + CKD group than those in the other groups and statistically significant. 

Diabetic parameters FBS (130.03 ± 52.07) and HbA1c levels (7.62 ± 1.46) were greater in the P + CKD group, which was statistically significant. Among renal parameters, serum creatinine (1.08 + 0.19) was greater in the CKD group, whereas eGFR (68.43 + 12.45) was significantly greater in the P + CKD group (Table 1).

Mean TNF-α levels were significantly greater in the P group (72.75 ± 22.86), followed by the P + CKD group (69.28 ± 18.07), when compared with the other groups (Table 1).

Mean CT values of *P.g*, *T.d* and *T.f* among the groups were lowest in the P + CKD group (24.33 + 2.39, 25.94 + 1.01, 27.90 + 2.02, respectively), followed by the P group. CT value was inversely proportional to the levels of the bacteria; hence, *P.g*, *T.d,* and *T.f* levels were significantly greater in the P+ CKD group, followed by the P group (Table 1).

When the red complex bacteria in all the groups were correlated with the rest of the variables, *P.g*, *T.d*, and *T.f* were positively correlated with all variables with a significant *p* value (Table 2). 

## 4. Discussion

Periodontitis, characterised by chronic inflammation of the periodontium, is caused by microbial pathogens affecting gingiva, periodontal ligament, and the alveolar bone; when left unchecked, it might result in compromised gingival health leading to tooth loss [18]. Early scientific evidence showed that disease progression occurs as a result of an increase in the colonisation of pathogenic subgingival flora, while recent studies have shifted their view on the exacerbated host immune response generated in response of this dysbiosis, thus leading to osteoclastic activity and bone loss [19]. The markers of inflammation produced during this complex interaction process between host and bacteria not only result in periodontal destruction, but also circulate from the oral biofilm into systemic circulation, thus triggering a chronic systemic immunoinflammatory response [20]. Hence, the association between periodontitis and CKD is credible, as systemic inflammation is an established risk factor of CKD, thus contributing to its progression and severity [21,22]. The current study investigates the inflammatory status and prevalence of red complex bacteria in the presence and absence of periodontitis with and without CKD with diabetes as one of the confounding factors.

The current study consisted of 120 subjects categorised in four groups: C (30 patients), P (30 patients), CKD (30 patients), and P + CKD (30 patients). Demographic variables, and diabetic, renal, inflammatory, and periodontal parameters were evaluated. Subgingival plaque samples were collected for the identification of *P. gingivalis*, *T. denticola,* and *T forsythia*. 

Demographic variables sex, body mass index (BMI), and socioeconomic status did not show any differences among groups and matched for further comparison among groups except for age. Age was significantly greater in the P + CKD group. Findings were accordance with those of studies by Mahendra et al., and Elango et al., who also matched demographic parameters while studying the association of periodontitis with pre-eclampsia and coronary heart disease, respectively [23,24]. This strongly establishes that age could be an important confounding factor for periodontitis and CKD. The literature suggests that, with increased age, patients are continuously exposed to their aetiological agents, thereby increasing the risk of periodontal destruction and contributing to the progression of other systemic inflammatory diseases such as CKD [25].

The comparison of periodontal parameters among groups showed that mean PI, PPD, GI, CAL, percentage of sites with PPD ≥ 5 mm, and CAL ≥ 3 mm were significantly greater in the P + CKD group. According to the World Workshop of Periodontology (2017), percentages of sites with PPD ≥ 5 mm and CAL ≥ 3 mm are two important factors in determining the severity of periodontitis. Similarly, a study by Borawski J et al. also found greater periodontal indices in patients with CKD [26]. A study by Desouza et al. stated greater periodontal indices in patients with CKD, suggesting the periodontal inflammation is a risk factor [27]. Past evidence also suggests that periodontitis consisting of inflammatory chronic-infection foci can disseminate through systemic circulation, thereby leading to the progression of systemic inflammatory disorders such as CKD [28,29]. Infection and inflammation mechanistically link systemic illness and periodontal disease. Since systemic inflammation is an established risk factor for CKD, the link between periodontitis and CKD is possible [21,22]. Comparing diabetic parameters such as FBS and HbA1c levels to those of the other groups showed that the levels were significantly greater in the P + CKD group. This was according to Chang et al., who also observed increased HbA1c levels in CKD patients with increased periodontal pocket depth [30]. A systemic review by Deschamps-Lenhardt et al. suggested that periodontitis contributes to the progression and severity of CKD, which may also worsen the diabetic status [14]. Attawood et al. suggested that periodontitis has both direct and indirect effects through diabetes on the incidence of CKD [10]. According to the longitudinal study of George Cindy et al., patients with Stage 3–5 CKD had greater HbA1c and fasting blood glucose levels, which was independent of other risk factors [31]. Hence, awareness about systemic comorbidities in periodontitis patients should be taken into consideration while associating with other systemic diseases. In conclusion, the above findings strengthen the association of a causal relationship between periodontitis and CKD progression, with diabetes as one of the major confounding factors, indicating that the second null hypothesis was validated.

Regarding renal parameters, serum creatinine was greater in the CKD groups than that in the other groups. A study by Naghsh and Narges et al. suggested that the serum level of creatinine (*p* = 0.02) had significant association between periodontitis severity and the increase in CAL in chronic kidney disease patients [32]. Aravindraj Velayutham et al. also reported increased serum creatinine levels in periodontitis cases [33]. Renal function and glomerular filtration rate are most commonly assessed by serum creatinine levels, which are usually elevated with impaired renal function. Similarly, eGFR was lower, suggesting compromised kidney function. This was in accordance with Iwasaki et al. who found that, in two years of a follow-up study, in Japanese elders, the greater risk of decreased renal function (reduction in eGFR) was associated with greater levels of periodontal inflammation [34]. Fisher et al. reported that host inflammatory burden is solely attributed to renal disease [35]. In CKD, there is a constant systemic inflammatory status, even in unknown aetiology; therefore, periodontitis-induced inflammatory response could synergise the inflammatory burden in these CKD patients [15]. Several studies reported that periodontal inflammation is prevalent in the predialytic stage of chronic renal patients [36,37]. 

TNF-α plays a vital function in progressing periodontitis by increasing the production of matrix metalloproteinases, thereby leading to periodontal destruction [38]. Impaired monocyte function may lead to changes in immune response to infectious agents and overproduce proinflammatory cytokines such as IL-1β, IL-6, and TNF-α [38]. In our study, mean TNF-α values were greater in the P and CKD + P groups in comparison with those of other groups. These results were similar to those of Niedzielska et al., who observed that periodontitis patients with CKD had a greater level of proinflammatory cytokines, including TNF-α. Ficek et al. reported higher levels of TNF-α in patients with acute renal failure (70 pg/mL), and significantly even greater levels in patients undergoing haemodialysis (216 pg/mL) [39]. Since periodontitis and CKD are based on an inflammatory milieu, periodontitis could play a role in CKD pathogenesis. Studies showed that inflammation is an important pathogenic factors in renal injury, and inflammatory markers such as TNF-α are positively correlated with CKD prevalence [40,41]. During the active phase of periodontitis, locally produced inflammatory cytokines such as interleukin-6 and tumour necrosis factor-α act systemically, which may lead to the progression of CKD [42]. Local inflammatory factors from affected gingiva enter the circulation, thereby increasing systemic inflammation and exacerbating progression. Chronic low-grade inflammation is a common phenomenon in patients with early stages (2 to 4) of CKD [42]. Different forms of chronic and acute inflammatory processes could induce an inflammatory response in the kidneys, leading to CKD progression [43]. 

Red complex bacteria remain a major risk for both periodontitis and CKD [44]. Studies support that red complex bacteria are at a greater percentile in patients with periodontal disease, which further aggravates kidney disease [45]. There is scientific evidence that suggests plausible mechanisms connecting periodontitis and CKD, which include periodontitis being a potential source of bacterial infection in CKD. The presence of periodontitis could aggravate or reactivate inborne bacterial infection in CKD [46,47,48]. 

In the present study, *P.g*, *T.f,* and *T.d* were greater in the P + CKD group than those in the other groups. Bastos et al. stated that prevailing periodontal bacteria such as *P.g*, *T.f,* and *T.d* are greater in CKD patients than in healthy individuals [49], thus validating the first null hypothesis. Fisher et al. evaluated the antibody titre against periodontal bacteria and found that 9% of CKD patients above 40 years of age had a greater antibody titre against P.g and other oral bacteria [50]. Our study also showed that *Pg*, *Td,* and *Tf* were significantly correlated with periodontal, diabetic, and renal parameters. This suggested that, with an increase in red complex bacterial levels, Egfr, serum creatinine, FBS, HbA1c, and TNF-α were elevated, thus suggesting strong correlation of the above variables in predialytic CKD patients (Table 2).

Schenkein et al., Dorn et al., and Takahashi et al. stated that *Pg*, *Td* and *A. actinomycetemcomitans* can invade endothelial cells, thus entering the circulation and resulting in bacteraemia [51,52,53]. This was further substantiated by studies by Takeuchi et al. (2011), Tomas et al. (2012), and Reyes et al. (2013) [54,55,56]. According to Shultis et al., periodontal bacteria leading to periodontal inflammation can lead to the progression of CKD [57].

The virulence factors of *P. gingivalis* contribute equally towards the pathogenesis of CKD. Most commonly, fimbriae fimA activates Toll-like receptors on macrophages leading to cAMP. The accumulation of cAMP-activated protein kinase A (PKA) destroys the bacterial nitric oxide synthase (iNOS) that is nuclear factor kappa B-dependent, thus contributing to phagocytosis of macrophages, which may trigger renal inflammation [58]. This further leads to the decreased production of nitric oxide by renal endothelial cells, thus contributing to the progression of renal fibrosis (Herrera et al., 2011) [59]. Hence, *P.gingivalis fimA* could be one of the contributing factors for the progression of CKD.

Similarly, *T.f* virulence factor, the serine proteinase inhibitor (serpin) protein, inhibits serine protease production from neutrophils. As a result, *T.f* is protected from the proteolytic activity of neutrophils, thereby eliciting the immune responses that may further affect renal function (Ksiazek et al., 2015) [58].

Chukkapalli et al. identified the genomic DNA of *P. gingivalis*, *F. nucleatum*, and *T. denticola* in the kidneys, thus showing the association of periodontal bacteria with compromised renal function [59]. 

Periodontal pathogens enter renal tissue and damage mesangial cells of the renal matrix, glomerulus, glomerular capillaries, and renal endothelium [13]. Hence, infected and inflamed periodontal pockets with highly organised subgingival Gram-negative biofilms, bacterial byproducts such as lipopolysaccharide (LPS), and proinflammatory cytokines function as a large reservoir disseminating through blood vessels and reaching distinct organs, thus leading to systemic inflammation [13]. Implementing gold standard nonsurgical periodontal therapy at the initial visit is thus necessary, as it aims at the removal of bacterial deposits. This facilitates reducing the process of inflammation, thereby preventing the occurrence of systemic infiltration. Though the therapy is considered to be the gold standard, eventual bacterial recolonisation after therapy is considered to be a major disadvantage [60]. Thus, the administration of adjunct agents that reduce or prevent bacterial recolonisation has gained importance, as they not only reduce bacterial load but also reverse it to a healthy state in optimal conditions. Owing to this field, probiotics in recent years have been in the limelight, as agents contain live microorganisms that, when administered in required amounts, offer a healthy benefit to the host. These probiotics exhibit an immunomodulant effect on periodontopathogens, preventing bacterial growth by numerous mechanisms. Though these probiotics have an adequate number of advantages, the safety margin of live microorganism administration remains a controversial question, especially when administered in immunocompromised individuals. Hence, newer modifications of these probiotics, such as parabiotics that contain inactive microbial cells to initiate the immune modulatory response, and postbiotics that contain microorganisms released during the metabolic activity of the same microorganism itself, are receiving attention in treating periodontitis [61]. 

Overall, this study shows telescopic knowledge on the periodontal inflammation and its influence on chronic kidney disease. Red complex bacteria and TNF-α levels are significantly associated with the aetiopathogenesis or aggravation of periodontitis and CKD. The study confirmed that the greater prevalence of red complex bacteria and TNF-α in CKD patients, with diabetes as one of the confounding factors, has emerged as one of the major contributors for the progression of periodontitis, which paves the way for perirenal continuum. 

This is the first study to assess bacterial status, and inflammatory, diabetic and renal biomarkers in an Indian population diagnosed with predialytic status with and without periodontitis, with diabetes as a confounding factor. 

A limitation of the study was that it lacked intervention. Further interventional longitudinal prospective studies are required to strengthen the role of periodontal management such as SRP, as one of the effective treatment modalities in controlling bacterial load and periodontal inflammation, thereby reducing the risk for impaired renal function in CKD patients.

## 5. Conclusions

Red complex bacteria are prevalent in the periodontitis and predialysis CKD patients when compared with those in the other groups, developing a different pathogenic mechanism interlinking the risk of CKD and periodontitis. This proposed link was further reinforced by the elevation of periodontal parameters and TNF-α levels, which substantiate the relationship between periodontal disease and progression of CKD, with diabetes as one of the confounding factors. Routine dental check-ups, providing gold-standard nonsurgical therapy (scaling and root planing), and the administration of evolving adjuncts such as parabiotics, probiotics, and postbiotics can be more effective in reducing oral inflammation and the risk of its systemic spread by reducing the bacterial load of periodontal-disease-causing pathogens. 

## Figures and Tables

**Figure 1 biology-11-00451-f001:**
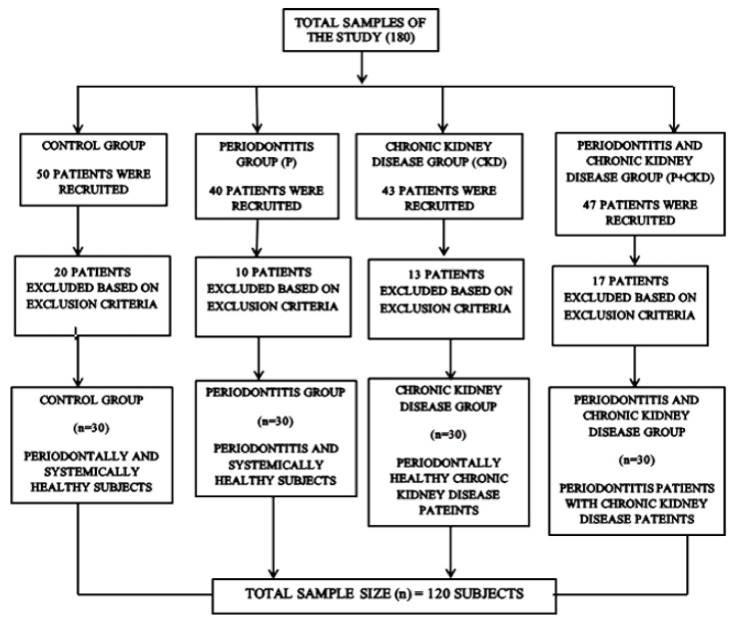
Study flowchart.

**Table 1 biology-11-00451-t001:** Comparison of demographic variables, renal, periodontal parameters and levels of red complex bacteria among groups.

Variable	C	P	CKD	P + CKD	ANOVA	*p* Value
Age (mean + SD)	37.63 + 10.26	54.03 + 9.10	59.27 + 10.90	61.47 + 10.99	32.56	0.004 *
Male (%)	22.2	20.6	27	30.2	0.39	0.173 (NS)
BMI	25.70 + 3.48	26.39 + 4.39	25.81 + 3.79	26.48 + 4.81	0.272	0.846 (NS)
Socioeconomic status	18,084 ± 5020	19,059 ± 4085	19,076± 4097	19,069 ± 4084	0.546	0.3 (NS)
Plaque index	0.79 + 0.29	1.79 + 0.22	0.88 + 0.29	2.27 + 0.23	225.748	0.001 *
Gingival index	0.91 + 0.34	1.92 + 0.35	1.05 + 0.30	2.42 + 0.18	172.28	0.001 *
PPD	1.33 + 0.17	2.77 + 0.27	1.29 + 0.09	3.18 + 0.24	675.859	0.002 *
Site percentage with PPD ≥ 5 mm	0 + 0	12.03 + 9.29	0 + 0	21.87 + 7.49	94.276	0.003 *
CAL	0 + 0	0.72 + 0.26	0 + 0	1.14 + 0.48	126.427	0.001 *
Site percentage with CAL ≥ 3 mm	0 + 0	19.36 + 7.72	0 + 0	30.43 + 11.00	150.83	0.00 *
FBS	89.43 ± 9.72	111.17 ± 48.08	103.73 ± 41.57	130.03 ± 52.07	5.008	0.003 *
HBA1C	5.28 ± 0.22	5.97 ± 1.16	6.01 ± 1.22	7.62 ± 1.46	7.439	0.000 *
Serum creatinine	0.80 + 0.14	0.72 + 0.11	1.20 + 0.43	1.08 + 0.19	25.20	0.00 *
eGFR	107.67 + 14.81	101 + 7.10	64.77 + 18.94	68.43 + 12.45	74.154	0.005 *
TNF-α	53.06 ± 25.51	72.75 ± 22.86	63.05 ± 29.90	69.28 ± 18.07	3.743	0.013
*P.g*	29.77 + 1.20	25.99 + 2.09	27 + 1.54	24.33 + 2.39	44.662	0.001 *
*T.d*	28.58 + 1.28	26.1 + 1.67	27.10 + 1.72	25.94 + 1.01	21.047	0.003 *
*T.f*	31.33 + 1.27	29.52 + 2.46	29.30 + 1.60	27.90 + 2.02	17.717	0.002 *

C, control group; P, periodontitis group; CKD, chronic kidney disease; P + CKD, periodontitis with chronic kidney disease; NS, not significant; *****, significant; *p* value ≤ 0.05 was considered to be significant.

**Table 2 biology-11-00451-t002:** Pearson correlation of red complex bacteria with periodontal, diabetic, and renal parameters.

	*P.g* CT Value		*T.d* CT Value		*T.f* CT Value	
	Pearson Correlation	*p* Value	Pearson Correlation	*p* Value	Pearson Correlation	*p* Value
PI score	−0.616	0.001 *	−0.455	0.004 *	−0.433	0.00 *
GI score	−0.571	0.003 *	−0.467	0.002 *	−0.432	0.002 *
TNF-A value (Pg/mL)	−0.244	0.007 *	−0.223	0.014 *	−0.200	0.03 *
Mean probing pocket depth (Mm)	−0.567	0.004 *	−0.473	0.001 *	−0.396	0.001 *
Mean clinical attachment loss	−0.504	0.002 *	−0.407	0.003 *	−0.361	0.003 *
E GFR value	0.479	0.000 *	.271	0.003 *	.473	0.00 *
Fasting blood sugar value (Mg/dL)	−0.197	0.031 *	−0.182	0.046 *	−0.208	0.02 *
Glycated haemoglobin score (%)	−0.243	0.008 *	−0.193	0.034 *	−0.267	0.00 *

CKD, chronic kidney disease; NS, not significant; *****, significant; *p* value ≤ 0.05 was considered to be significant.

## Data Availability

Not applicable.

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
