# Peer review of "Impact of Red Complex Bacteria and TNF-α Levels on the Diabetic and Renal Status of Chronic Kidney Disease Patients in the Presence and Absence of Periodontitis"

_biology, 2022, doi:10.3390/biology11030451_

Round 1

Reviewer 1 Report

Manuscript of considerable interest, due to the multidisciplinary approach in the management of the periodontal patient with systemic diseases, need for revisions.

Abstracts and keywords written correctly.

Introduction, to add the new classification of periodontal disease and on the basis of this the type of patients enrolled.

Materials and methods, how was the sample size calculated?

When was the plaque removal done?

How were local risk factors subsequently removed?

Results: By arranging the data in the table, they are very confusing and emphasizing the statistically significant data so that the reader can read them easily.
Make the flow chart in high resolution.

Discussions: implementing a proactive approach for the maintenance of eubiosis of the oral cavity, for future research based on para probiotics, probiotics and post biotics for the reduction of the bacterial load of the orange and red complex, as studied by the research group of Scribante et al. and the evaluation of periodontal therapy on oxidative stress by the research group of Marconcini et al.

Conclusions, to reformulate the modulation of the bacterial load of the red complex on the basis of probiotics

Biobliography: references from the discussion to be added

Author Response

REPLY TO THE COMMENTS OF THE REVIEWERS

Reviewer 1:

Manuscript of considerable interest, due to the multidisciplinary approach in the management of the periodontal patient with systemic diseases, need for revisions.

Abstracts and keywords written correctly.

Question 1: Introduction, to add the new classification of periodontal disease and on the basis of this the type of patients enrolled.

Answer: The sentence has been added and highlighted as per reviewer’s suggestion. (Line number: 83-90)

Question 2: Materials and methods, how was the sample size calculated?

Answer: The sentence has been added as per reviewer’s suggestion and highlighted for the reference. (Line number: 117-118)

Question 3: When was the plaque removal done?

Answer: The time of plaque collection has been added and highlighted in the materials and methods part of the manuscript as per reviewer’s suggestion. (Line number: 179)

Question 4: How local risk factors were subsequently removed?

Answer: The sentence has been added and highlighted in materials and method section as per reviewer’s suggestion. (Line number: 184,185, 186)

Question 5: Results: By arranging the data in the table, they are very confusing and emphasizing the statistically significant data so that the reader can read them easily.
Make the flow chart in high resolution.

Answer: The table has been simplified and the statistically significant values have been highlighted in bold font as per reviewer’s suggestion .

Question 6: Discussions: implementing a proactive approach for the maintenance of eubiosis of the oral cavity, for future research based on para probiotics, probiotics and post biotics for the reduction of the bacterial load of the orange and red complex, as studied by the research group of Scribante et al. and the evaluation of periodontal therapy on oxidative stress by the research group of Marconcini et al.

Answer: The discussion part has been modified and highlighted as per reviewer’s suggestion. (Line number: 529-546)

Question 7: Conclusions, to reformulate the modulation of the bacterial load of the red complex on the basis of probiotics

Answer: The conclusion part has been modified and highlighted as per reviewer’s suggestion. (Line number: 571- 575)

Question 8: Biobliography: references from the discussion to be added

Answer: The reference number of 60,61 of the discussion part has been added to bibliography as per reviewer’s suggestion. (Line number: 756-761)

Reviewer 2:

Dear Authors,

Thank you for your work which I found very interesting. Here are some comments to improve its quality:

ABSTRACT

Question 1: The abstract should be shorter (at maximum 250/300 words) and written without captions, as requested by the journal.

Answer: The abstract has been modified and the word count has been reduced as per reviewer’s suggestion. (Line number: 39-62)

INTRODUCTION

Question 2: Please specify periodontitis according to the latest classification (2017)

Answer: The latest classification of periodontitis has been specified as per reviewer’s suggestions. (Line number: 82-90)

Question 3: At the end of the introduction, please specify the statistical null hypothesis

Answer: The null hypothesis has been added at the end of introduction and highlighted as per reviewer’s suggestion. (Line number: 105-110)

MATERIALS AND METHODS

Question 4: Subsections should be listed with numbers (e.g.; 2.1.; 2.2; 2.3, ecc.)

Answer: The numbering of the subsections has been added and highlighted as per reviewer’s suggestion.

Question 5: The statistical analysis section should be reported at the end of the materials and methods section

Answer: The word has been removed and added at the end of materials and methods as per reviewer’s suggestion. (Line number:247)

Question 6: Was a sample size calculation performed?

 Answer: The sample size calculation has been performed and the lines are added and highlighted as per reviewer’s suggestion. (Line number: 117, 118)

RESULTS

Question 7: The flow chart, whose quality must be improved, should be reported at the beginning of the section

Answer: The image quality has been improved and added as per reviewer’s suggestion.

Question 8: Tables are not so clear, they should be reformulated

Answer: The tables have been reformulated as per reviewer’s suggestion.

DISCUSSION

Question 9: Please remove the subsection title

Answer: The subsection title has been removed as per reviewer’s suggestion.

Question 10: Please specify whether the statistical null hypothesis has been accepted or not.I think that the discussion is well constructed, however it could be interesting to add a specific paragraph discussing the recent introduction of probiotics for the non-surgical periodontal treatment. You could refer to the following article assessing clinical and microbiological improvements after probiotics intake: https://pubmed.ncbi.nlm.nih.gov/33383903/

Answer: The lines are added as per reviewer’s suggestion and has been highlighted. (Line number: 443, 444, 493)

Reviewer 2 Report

Dear Authors,

thank you for your work which I found very interesting. Here are some comments to improve its quality:

ABSTRACT

  • The abstract should be shorter (at maximum 250/300 words) and written without captions, as requested by the journal.

INTRODUCTION

  • Please specify periodontitis according to the latest classification (2017)
  • At the end of the introduction, please specify the statistical null hypothesis

MATERIALS AND METHODS

  • Subsections should be listed with numbers (e.g.; 2.1.; 2.2; 2.3, ecc.)
  • Line 106: the statistical analysis section should be reported at the end of the materials and methods section
  • Was a sample size calculation performed?

RESULTS

  • The flow chart, whose quality must be improved, should be reported at the beginning of the section
  • Tables are not so clear, they should be reformulated

DISCUSSION

  • Line 379: please remove the subsection title
  • Please specify whether the statistical null hypothesis has been accepted or not.
  • I think that the discussion is well constructed, however it could be interesting to add a specific paragraph discussing the recent introduction of probiotics for the non-surgical periodontal treatment. You could refer to the following article assessing clinical and microbiological improvements after probiotics intake: https://pubmed.ncbi.nlm.nih.gov/33383903/

Thank you again.

The Reviewer

Author Response

(The authors gave the same response as above.)

Round 2

Reviewer 1 Report

The manuscript has been properly revised